# Cooperating with machines

Jacob W. Crandall [1], Mayada Oudah [2], Tennom[3], Fatimah Ishowo-Oloko [2], Sherief Abdallah [4,5], Jean-François Bonnefon[6], Manuel Cebrian[7], Azim Shariff[8], Michael A. Goodrich [1] & Iyad Rahwan [7,9]

Since Alan Turing envisioned artificial intelligence, technical progress has often been measured by the ability to defeat humans in zero-sum encounters (e.g., Chess, Poker, or Go). Less attention has been given to scenarios in which human–machine cooperation is beneficial but non-trivial, such as scenarios in which human and machine preferences are neither fully aligned nor fully in conflict. Cooperation does not require sheer computational power, but instead is facilitated by intuition, cultural norms, emotions, signals, and pre-evolved dispositions. Here, we develop an algorithm that combines a state-of-the-art reinforcement-learning algorithm with mechanisms for signaling. We show that this algorithm can cooperate with people and other algorithms at levels that rival human cooperation in a variety of two-player repeated stochastic games. These results indicate that general human–machine cooperation is achievable using a non-trivial, but ultimately simple, set of algorithmic mechanisms.

[1] Computer Science Department, Brigham Young University, 3361 TMCB, Provo, UT 84602, USA. [2] Khalifa University of Science and Technology, Masdar Institute, P.O. Box 54224, Abu Dhabi, United Arab Emirates. [3] UVA Digital Himalaya Project, University of Virginia, Charlottesville, VA 22904, USA. [4] British University in Dubai, Dubai, United Arab Emirates. [5] School of Informatics, University of Edinburgh, Edinburgh EH8 9AB, UK. [6] Toulouse School of Economics (TSM-Research), Centre National de la Recherche Scientifique, University of Toulouse Capitole, Toulouse 31015, France. [7] The Media Lab, Massachusetts Institute of Technology, Cambridge, MA 02139, USA. [8] Department of Psychology and Social Behavior, University of California, Irvine, CA 92697, USA. [9] Institute for Data, Systems and Society, Massachusetts Institute of Technology, 77 Massachusetts Avenue, Cambridge, MA 02139, USA. Correspondence and requests for materials should be addressed to J.W.C. (email: crandall@cs.byu.edu) or to I.R. (email: irahwan@mit.edu)

The emergence of driverless cars, autonomous trading algorithms, and autonomous drone technologies highlight a larger trend in which artificial intelligence (AI) is enabling machines to autonomously carry out complex tasks on behalf of their human stakeholders. To effectively represent their stakeholders in many tasks, these autonomous machines must interact with other people and machines that do not fully share the same goals and preferences. While the majority of AI milestones have focused on developing human-level wherewithal to compete with people[1–6] or to interact with people as teammates that share a common goal[7–9], many scenarios in which AI must interact with people and other machines are neither zero-sum nor common-interest interactions. As such, AI must also have the ability to cooperate even in the midst of conflicting interests and threats of being exploited. Our goal in this paper is to better understand how to build AI algorithms that cooperate with people and other machines at levels that rival human cooperation in arbitrary long-term relationships modeled as repeated stochastic games (RSGs).

Algorithms capable of forming cooperative long-term relationships with people and other machines in arbitrary repeated games are not easy to come by. A successful algorithm should possess several properties. First, it must not be domain-specific—it must have superior performance in a wide variety of scenarios (generality). Second, the algorithm must learn to establish effective relationships with people and machines without prior knowledge of associates' behaviors (flexibility). To do this, it must be able to deter potentially exploitative behavior from its partner and, when beneficial, determine how to elicit cooperation from a (potentially distrustful) partner who might be disinclined to cooperate. Third, when associating with people, the algorithm must learn effective behavior within very short timescales—i.e., within only a few rounds of interaction (learning speed). These requirements create many technical challenges (Supplementary Note 2), including the need to deal with adaptive partners who may also be learning[10,11] and the need to reason over multiple, potentially infinite, equilibria solutions within the large strategy spaces inherent of repeated games. The sum of these challenges often causes AI algorithms to fail to cooperate[12], even when doing so would be beneficial to the algorithm's long-term payoffs.

Human cooperation does not require sheer computational power, but is rather facilitated by intuition[13], cultural norms[14,15], emotions and signals[16,17], and pre-evolved dispositions toward cooperation[18]. Of particular note, cheap talk (i.e., costless, non-binding signals) has been shown to lead to greater human cooperation in repeated interactions[19,20]. These prior works suggest that machines may also rely on such mechanisms, both to deal with the computational complexities of the problem at hand and to develop shared representations with people[21–24]. However, it has remained unclear how autonomous machines can emulate these mechanisms in a way that supports generality, flexibility, and fast learning speeds.

The primary contribution of this work is threefold. First, we conduct an extensive comparison of existing algorithms for repeated games. Second, we develop and analyze a learning algorithm that couples a state-of-the-art machine-learning algorithm (the highest performing algorithm in our comparisons of algorithms) with mechanisms for generating and responding to signals at levels conducive to human understanding. Finally, via extensive simulations and user studies, we show that this learning algorithm learns to establish and maintain effective relationships with people and other machines in a wide variety of RSGs at levels that rival human cooperation, a feat not achieved by prior algorithms. In so doing, we investigate the algorithmic mechanisms that are responsible for the algorithm's success. These results are summarized and discussed in the next section, and given in full detail in the Supplementary Notes.

## Results

**Evaluating the state-of-the-art**. Over the last several decades, algorithms for generating strategic behavior in repeated games have been developed in many disciplines, including economics, evolutionary biology, and the AI and machine-learning communities[10–12][25–36]. To evaluate the ability of these algorithms to forge successful relationships, we selected 25 representative algorithms from these fields, including classical algorithms such as (generalized) Generous Tit-for-Tat (i.e., Godfather[36]) and win-stay-lose-shift (WSLS)[27], evolutionarily evolved memory-one and memory-two stochastic strategies[35], machine-learning algorithms (including reinforcement-learning), belief-based algorithms[25], and expert algorithms[32,37]. Implementation details used in our evaluation for each of these algorithms are given in Supplementary Note 3.

We compared these algorithms with respect to six performance metrics at three different game lengths: 100-, 1000-, and 50,000-round games. The first metric was the Round-Robin average, which was calculated across all games and partner algorithms (though we also considered subsets of games and partner algorithms). This metric tests an algorithm's overall ability to forge profitable relationships across a range of potential partners. Second, we computed the best score, which is the percent of algorithms against which an algorithm had the highest average payoff compared to all 25 algorithms. This metric evaluates how often an algorithm would be the desired choice, given knowledge of the algorithm used by one's partner. The third metric was the worst-case score, which is the lowest relative score obtained by the algorithm. This metric addresses the ability of an algorithm to bound its losses. Finally, the last three metrics are designed to evaluate the robustness of algorithms to different populations. These metrics included the usage rate of the algorithms over 10,000 generations of the replicator dynamic[38], and two forms of elimination tournaments[35]. Formal definitions of these metrics and methods for ranking the algorithms with respect to them are provided in Supplementary Note 3.

As far as we are aware, none of the selected algorithms had previously been evaluated in this extensive set of games played against so many different kinds of partners, and against all of these performance metrics. Hence, this evaluation illustrates how well these algorithms generalize in two-player normal-form games, rather than being fine-tuned for specific scenarios.

Results of the evaluation are summarized in Table 1, with more detailed results and analysis appearing in Supplementary Note 3. We make two high-level observations here. First, it is interesting which algorithms were less successful in these evaluations. For instance, while Generalized Tit-for-Tat (gTFT), WSLS, and memory-one and memory-two stochastic strategies (e.g., Mem-1 and Mem-2) are successful in prisoner's dilemmas, they are not consistently effective across the full set of $2 \times 2$ games. These algorithms are particularly ineffective in longer interactions, as they do not effectively adapt to their partner's behavior. Additionally, algorithms that minimize regret (e.g., Exp3[29], GIGA-WoLF[39], and WMA[40]), which is the central component of world-champion computer poker algorithms[4,5], also performed poorly.

Second, while many algorithms had high performance with respect to some measure, only S++[37] was a top-performing algorithm across all metrics at all game lengths. Furthermore, results reported in Supplementary Note 3 show that it maintained this high performance in each class of game and when associating with each class of algorithm. S++ learns to cooperate with like-minded partners, exploit weaker competition, and bound its worst-case performance (Fig. 1a). Perhaps most importantly,

**Table 1 Summary results for our comparison of algorithms**

| Algorithm | Round-Robin average | % Best score | Worst-case score | Replicator dynamic | Group-1 Tourney | Group-2 Tourney | Rank summary min–mean–max |
|---|---|---|---|---|---|---|---|
| S++ | 1, 1, 1 | 2, 1, 2 | 1, 1, 1 | 1, 1, 1 | 1, 1, 2 | 1, 1, 1 | 1–1.2–2 |
| Manipulator | 3, 2, 3 | 4, 3, 8 | 5, 2, 4 | 6, 4, 3 | 5, 3, 3 | 5, 2, 2 | 2–3.7–8 |
| Bully | 3, 2, 1 | 3, 2, 1 | 7, 13, 20 | 7, 3, 2 | 6, 2, 1 | 6, 3, 5 | 1–4.8–20 |
| S++/simple | 5, 4, 4 | 8, 5, 9 | 4, 6, 10 | 10, 2, 6 | 8, 4, 6 | 9, 4, 6 | 2–6.1–10 |
| S | 5, 5, 8 | 6, 7, 10 | 3, 3, 8 | 5, 5, 8 | 7, 5, 9 | 7, 5, 9 | 3–6.4–10 |
| Fict. play | 2, 8, 14 | 1, 6, 10 | 2, 8, 16 | 3, 12, 15 | 2, 8, 12 | 4, 9, 14 | 1–8.1–16 |
| MBRL-1 | 6, 6, 10 | 5, 4, 7 | 8, 7, 14 | 11, 11, 13 | 9, 7, 10 | 8, 7, 10 | 4–8.5–14 |
| EEE | 11, 8, 7 | 14, 9, 5 | 9, 4, 2 | 14, 10, 9 | 13, 9, 8 | 13, 10, 8 | 2–9.1–14 |
| MBRL-2 | 14, 5, 5 | 13, 8, 6 | 19, 5, 3 | 18, 9, 4 | 18, 6, 5 | 18, 6, 4 | 3–9.2–19 |
| Mem-1 | 6, 9, 13 | 7, 10, 21 | 6, 9, 17 | 2, 6, 10 | 3, 10, 17 | 2, 8, 15 | 2–9.5–21 |
| M-Qubed | 14, 20, 4 | 15, 20, 3 | 15, 19, 5 | 17, 19, 5 | 17, 21, 4 | 16, 21, 3 | 3–13.2–21 |
| Mem-2 | 9, 11, 20 | 9, 11, 22 | 13, 17, 22 | 4, 13, 19 | 4, 13, 25 | 3, 12, 20 | 3–13.7–25 |
| Manip-Gf | 11, 11, 21 | 12, 12, 19 | 12, 11, 19 | 9, 7, 20 | 12, 14, 20 | 11, 13, 21 | 7–14.2–21 |
| WoLF-PHC | 17, 11, 13 | 18, 14, 14 | 18, 14, 18 | 16, 14, 14 | 16, 11, 11 | 15, 11, 11 | 11–14.2–18 |
| QL | 17, 17, 7 | 19, 19, 4 | 17, 18, 7 | 19, 18, 7 | 19, 20, 7 | 19, 18, 7 | 4–14.4–20 |
| gTFT | 11, 14, 22 | 11, 15, 20 | 11, 16, 23 | 8, 8, 22 | 10, 16, 21 | 10, 15, 22 | 8–15.3–23 |
| EEE/simple | 20, 15, 11 | 20, 17, 12 | 20, 10, 9 | 20, 16, 11 | 24, 15, 14 | 20, 16, 13 | 9–15.7–24 |
| Exp3 | 19, 23, 11 | 16, 23, 15 | 16, 23, 6 | 15, 23, 12 | 15, 25, 13 | 17, 25, 12 | 6–17.2–25 |
| CJAL | 24, 14, 14 | 25, 14, 13 | 24, 12, 15 | 24, 17, 16 | 20, 12, 16 | 22, 14, 16 | 12–17.3–25 |
| WSLS | 9, 17, 24 | 10, 16, 24 | 10, 20, 24 | 12, 20, 24 | 11, 17, 24 | 12, 17, 25 | 9–17.6–25 |
| GIGA-WoLF | 14, 19, 23 | 17, 18, 23 | 14, 15, 21 | 13, 15, 23 | 14, 18, 22 | 14, 19, 23 | 13–18.1–23 |
| WMA | 21, 21, 15 | 21, 21, 16 | 22, 21, 12 | 22, 21, 17 | 21, 19, 15 | 23, 20, 17 | 12–19.2–23 |
| Stoch. FP | 21, 21, 15 | 22, 22, 17 | 23, 22, 11 | 23, 22, 18 | 25, 24, 18 | 25, 22, 18 | 11–20.5–25 |
| Exp3/simple | 21, 24, 16 | 23, 24, 18 | 21, 24, 13 | 21, 24, 21 | 22, 22, 19 | 21, 23, 19 | 13–20.9–24 |
| Random | 24, 25, 25 | 24, 25, 25 | 25, 25, 25 | 25, 25, 25 | 23, 23, 23 | 24, 24, 24 | 23–24.4–25 |

This summary gives the relative rank of each algorithm with respect to each of the six performance metrics we considered, at each game length. A lower rank indicates higher performance. For each metric, the algorithms are ranked in 100-round, 1000-round, and 50,000-round games, respectively. For example, the 3-tuple 3, 2, 1 indicates the algorithm was ranked 3rd, 2nd, and 1st in 100, 1000, and 50,000-round games, respectively. More detailed results and explanations are given in Supplementary Note 3

whereas many machine-learning algorithms do not forge cooperative relationships until after thousands of rounds of interaction (if at all) (e.g. ref. [41]), S++ tends to do so within relatively few rounds of interaction (Fig. 1b), likely fast enough to support interactions with people.

Given S++'s consistent success when interacting with other algorithms, we also evaluated its ability to forge cooperative relationships with people via a user study in which we paired S++ and MBRL-1, a model-based reinforcement-learning algorithm[42], with people in four repeated games. The results of this user study, summarized in Fig. 2, show that while S++ establishes cooperative relationships with copies of itself, it does not consistently forge cooperative relationships with people. Across the four games, pairings consisting of S++ and a human played the mutually cooperative solution (i.e., the Nash bargaining solution) in <30% of rounds. Nevertheless, people likewise failed to consistently cooperate with each other in these games. See Supplementary Note 5 for additional details and results.

In summary, none of the 25 existing algorithms we evaluated establishes effective long-term relationships with both people and other algorithms.

**An algorithm that cooperates with people and other machines.** Prior work has shown that humans rely on costless, non-binding signals (called cheap talk) to establish cooperative relationships in repeated games[19,20]. Thus, we now consider scenarios that permit such communications. While traditional repeated games do not provide means for cheap talk, we consider a richer class of repeated games in which players can engage in cheap talk by sending messages at the beginning of each round. Consistent with prior work[20], we limited messages to a pre-determined set of speech acts.

While signaling via cheap talk comes naturally to humans, prior algorithms designed for repeated games are not equipped

with the ability to generate and respond to such signals. To be useful in establishing relationships with people in arbitrary scenarios, costless signals should be connected to actual behavior, should be communicated at a level conducive to human understanding, and should be generated by game-independent mechanisms. The ability to utilize such signals relates to the idea of explainable AI, which has recently become a grand challenge[43]. This challenge arises due to the fact that most machine-learning algorithms have low-level internal representations that are not easily understood or expressed at levels that are understandable to people, especially in arbitrary scenarios. As such, it is not obvious how machine-learning algorithms can, in addition to prescribing strategic behavior, generate and respond to costless signals at levels that people understand in arbitrary scenarios.

Unlike typical machine-learning algorithms, the internal structure of S++ provides a high-level representation of the algorithm's dynamic strategy that can be described in terms of the dynamics of the underlying experts. Since each expert encodes a high-level philosophy, S++ can be used to generate signals (i.e., cheap talk) that describe its intentionality. Speech acts from its partner can also be compared to its experts' philosophies to improve its expert-selection mechanism. In this way, we augmented S++ with a communication framework that gives it the ability to generate and respond to cheap talk.

The resulting new algorithm, dubbed S# (pronounced "S sharp"), is depicted in Fig. 3; details of S# are provided in Methods and Supplementary Note 4. In scenarios in which cheap talk is not possible, S# is identical to S++. When cheap talk is permitted, S# differs from S++ in that it generates cheap talk that corresponds to the high-level behavior and state of S++ and uses the signals spoken by its partner to alter which expert it chooses to follow in order to more easily coordinate behavior. Since self-play analysis indicated that both of these mechanisms help facilitate cooperative relationships (see Supplementary Note 4, in

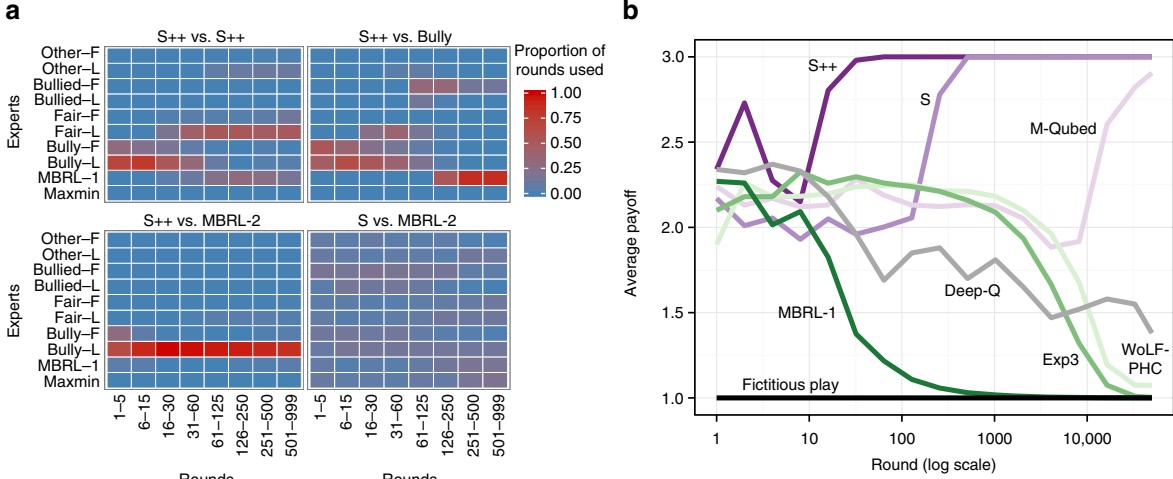

**Fig. 1** Illustrations of S++'s learning dynamics. **a** An illustration of S++'s learning dynamics in Chicken, averaged over 50 trials. For ease of exposition, S++'s experts are categorized into groups (see Supplementary Note 3 for details). Top-left: When (unknowingly) paired with another agent that uses S++, S++ initially tries to bully its partner, but then switches to fair, cooperative experts when attempts to exploit its partner are unsuccessful. Top-right: When paired with Bully, S++ learns the best response, which is to be bullied, achieved by playing experts MBRL-1, Bully-L, or Bully-F. Bottom-left: S++ quickly learns to play experts that bully MBRL-2, meaning that it receives higher payoffs than MBRL-2. Bottom-right: on the other hand, algorithm S does not learn to consistently bully MBRL-2, showing that S++'s pruning rule (see Eq. 1 in Methods) enables it to teach MBRL-2 to accept being bullied, thus producing high payoffs for S++. **b** The average per-round payoffs (averaged over 50 trials) of various machine-learning algorithms over time in self-play in a traditional (0-1-3-5)-prisoner's dilemma in which mutual cooperation produces a payoff of 3 and mutual defection produces a payoff of 1. Of the machine-learning algorithms we evaluated, only S++ quickly forms successful relationships with other algorithms across the set of 2 × 2 games

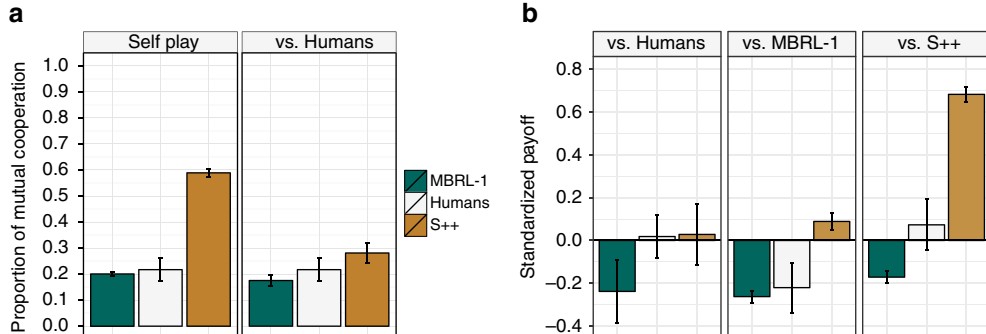

**Fig. 2** Results from an initial user study. In this study, participants were paired with other people, S++, and MBRL-1 in four different repeated normal-form games. Each game consisted of 50+ rounds. **a** The proportion of rounds in which both players cooperated when a player was paired with a partner of like type (self-play) and with a human. **b** The standardized payoff, computed as the standardized z-score, obtained by each algorithm when paired with each partner type. In both plots, error bars show the standard error of the mean. These plots show that while S++ learns to cooperate with a copy of itself, it fails to consistently forge cooperative relationships with people. There were some variations across games. Details about the user study and results are provided in Supplementary Note 5

particular Supplementary Figure 12), we anticipated that, in combination with S++'s ability to learn effective behavior when paired with both cooperative and devious partners, S#'s signaling mechanisms could be the impetus for consistently forging cooperative relationships with people.

We conducted a series of three user studies involving 220 participants, who played in a total of 472 repeated games, to determine the ability of S# to forge cooperative relationships with people. The full details of these studies are provided in Supplementary Notes 5–7. We report representative results from the final study, in which participants played three representative repeated games (drawn from distinct payoff families; see Methods and Supplementary Note 2) via a computer interface that hid their partner's identity. In some conditions, players could engage in cheap talk by sending messages at the beginning of each round via the computer interface.

The proportion of mutual cooperation achieved by human–human, human–S#, and S#–S# pairings are shown in Fig. 4. When cheap talk was not permitted, human–human and human–S# pairings did not frequently result in cooperative relationships. However, across all three games, the presence of cheap talk doubled the proportion of mutual cooperation experienced by these two pairings. Thus, like people, S# used cheap talk to greatly enhance its ability to forge cooperative relationships with humans. Furthermore, while S#'s speech profile was distinct from that of humans (Fig. 5a), subjective post-interaction assessments indicate that S# used cheap talk to promote cooperation as effectively as people (Fig. 5b). In fact,

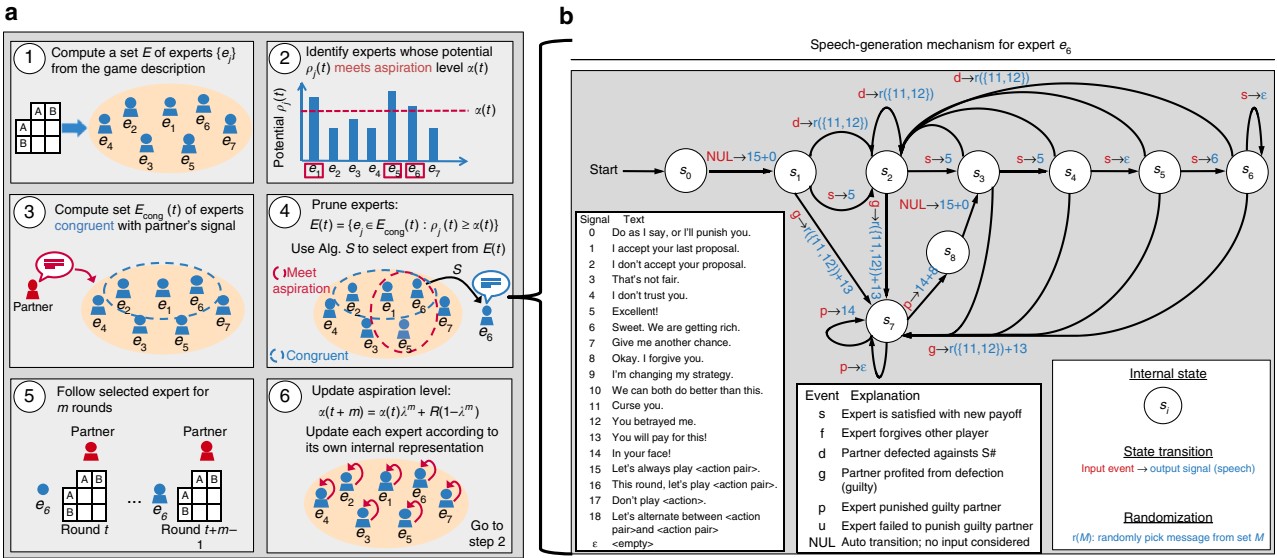

**Fig. 3** An overview of S#. S# extends S++[37] with the ability to generate and respond to cheap talk. **a** S#'s algorithmic steps. Prior to any interaction, S# uses the description of the game to compute a set E of expert strategies (step 1). Each expert encodes a strategy or learning algorithm that defines both behavior and speech acts (cheap talk) over all game states. In step 2, S# computes the potential, or highest expected utility, of each expert in E. The potentials are then compared to an aspiration level $\alpha(t)$, which encodes the average per-round payoff that the algorithm believes is achievable, to determine a set of experts that could potentially meet its aspiration. In step 3, S# determines that experts carry out plans that are congruent with its partner's last proposed plan. Next, in step 4, S# selects an expert, using algorithm S[34,60], from among those experts that both potentially meet its aspiration (step 2) and are congruent with its partner's latest proposal (step 3). If E(t) is empty, S# selects its expert from among those experts that meet its aspiration (step 2). The currently selected expert generates signals (speech from a pre-determined set of speech acts) based on its game-generic state machine **b**. In step 5, S# follows the strategy dictated by the selected expert for m rounds of the repeated game. Finally, in step 6, S# updates its aspiration level based on the average reward R it has received over the last m rounds of the game. It also updates its experts according to each expert's internal representation. It then returns to step 2 and repeats the process for the duration of the repeated game. **b** An example speech-generation mechanism for an expert that seeks to teach its partner to play a fair, pareto-optimal strategy. For each expert, speech is generated using a state machine (specifically, a Mealy machine[70]), in which the algorithm's states are the nodes, algorithmic events create transitions between nodes, and speech acts define the outputs

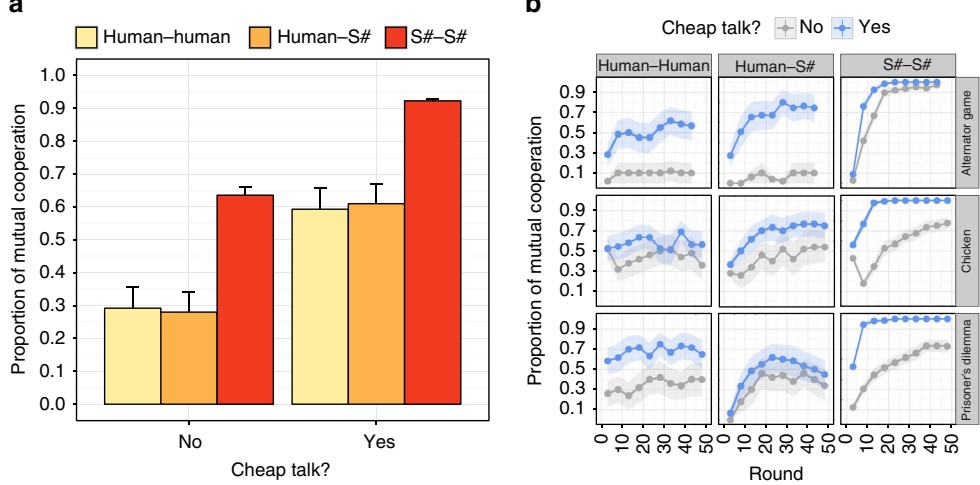

**Fig. 4** Proportion of mutual cooperation achieved in the culminating user study. In this study, 66 volunteer participants were paired with each other and S# in three representative games (Chicken, Alternator Game, and Prisoner's Dilemma). Results are shown for when cheap talk was both permitted and not permitted. Note that S# is identical to S++ when cheap talk is not permitted. **a** The proportion of rounds in which both players cooperated over all rounds and all games. **b** The proportion of time in which both players cooperated over all games. Bars and lines show average values over all trials, while error bars and ribbons show the standard error of the mean. A full statistical analysis confirming the observations shown in this figure are provided in Supplementary Note 7

many participants were unable to distinguish S# from a human player (Fig. 5c).

In addition to demonstrating S#'s ability to form cooperative relationships with people, Fig. 4 also shows that S#–S# pairings were more successful than either human–human or human–S#

pairings. In fact, S++–S++ pairings (i.e., S#–S# pairings without the ability to communicate) achieved cooperative relationships as frequently as human–human and human–S# pairings that were allowed to communicate via cheap talk (Fig. 4a). Given the ability to communicate, S#–S# pairings were far more consistent in

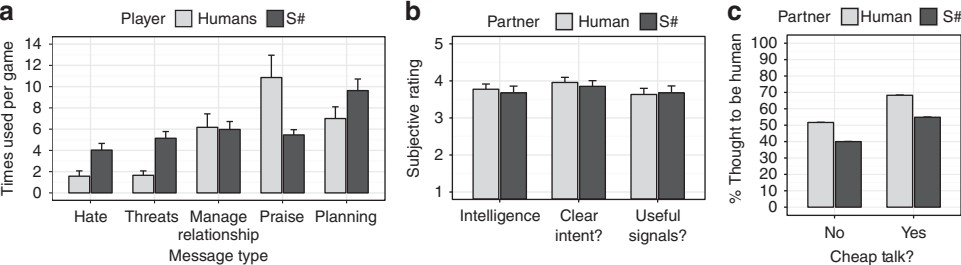

**Fig. 5** Explanatory results from the culminating user study. **a** The speech profiles of participants and S#, which shows the average number of times that people and S# used messages of each type (see Supplementary Note 7 for message classification) over the course of an interaction when paired with people across all games. Statistical tests confirm that S# sent significantly more Hate and Threat messages, while people sent more praise messages. **b** Results of three post-experiment questions for subjects that experienced the condition in which cheap talk was permitted. Participants rated the intelligence of their partner (Intelligence), the clarity of their partner's intentions (Clear intent?), and the usefulness of the communication between them and their partner (Useful signals?). Answers were given on a 5-point Likert scale. Statistical analysis did not detect any significant differences in the way users rated S# and human partners. **c** The percentage of time that human participants and S# were thought to be human by their partner. Statistical analysis did not detect any difference in the rates at which S# and humans were judged to be human. Further details along with the statistical analysis for each of these results are provided in Supplementary Note 7

forging cooperative relationships than both human–human and human–S# pairings.

Together, these results illustrate that, across the games studied, the combined behavioral and signaling strategies of S# were at least as effective as those of human players.

**Repeated stochastic games**. While the previous results appertained to normal-form games, S++ is also effective in more complex scenarios[44]. This facilitates an extension of S# to these more complex games. Because normal-form games capture the essence of the dilemmas faced by people in repeated interactions, they have been used to study cooperation for decades in the fields of behavioral economics[45], mathematical biology[46], psychology[47], sociology[48], computer science[36], and political science[26]. However, such scenarios abstract away some complexities of real-world interactions. Thus, we also consider the more general class of RSGs, which require players to reason over sequences of moves in each round, rather than the single-move rounds of normal-form games.

To generate and respond to signals in RSGs, S# uses the same mechanisms as it does in normal-form games with one exception. In normal-form games, joint plans for a round can be communicated by specifying a joint action. However, in RSGs, plans are often more complex as they involve a series of joint actions that would not typically be used in human communication. In such cases, S# instead communicates plans with higher-level terms such as "Let's cooperate" or "I get the higher payoff," and then relies on its partner to infer the specifics of the proposed joint strategy. See Methods for details.

Our results for RSGs are similar to those of normal-form games. While S++ does not typically forge effective relationships with people in these more complex scenarios, our results show that S#-, an early version of S# that generated cheap talk but did not respond to the cheap talk of others, is more successful at doing so. For example, Fig. 6 shows results for a turning-taking scenario in which two players must learn how to share a set of blocks. Like people, S#- used cheap talk to substantially increase its payoffs when associating with people in this game (Fig. 6b). Though S#- was limited by its inability to respond to the cheap talk of others (Supplementary Note 4; see, in particular, Supplementary Table 12), this result mirrors those we observed in normal-form games. (Supplementary Note 6 contains additional details and results.) This illustrates that S# can also be used in more complex scenarios to forge cooperative relationships with people.

**Distinguishing algorithmic mechanisms**. Why is S# so successful in forging cooperative relationships with both people and other algorithms? Are its algorithmic mechanisms fundamentally different from those of other algorithms for repeated games? We have identified three algorithmic mechanisms responsible for S#'s success. Clearly, Figs. 4, 5, 6 demonstrate that the first of these mechanisms is S#'s ability to generate and respond to relevant signals that people can understand, a trait not present in previous learning algorithms designed for repeated interactions. These signaling capabilities expand S#'s flexibility in that they also allow S# to more consistently forge cooperative relationships with people. Without this capability, it does not consistently do so. Figure 7a demonstrates one simple reason that this mechanism is so important: cheap talk helps both S# and humans to more quickly develop a pattern of mutual cooperation with their partners. Thus, the ability to generate and respond to signals at a level conducive to human understanding is a critical algorithmic mechanism.

Second, S# uses a rich set of expert strategies that includes a variety of equilibrium strategies and even a simple learning algorithm. While none of these individual experts has an overly complex representation (e.g., no expert remembers the full history of play), these experts are more sophisticated than those traditionally considered (though not explicitly excluded) in the discussion of expert algorithms[29,39,40]. This more sophisticated set of experts permits S# to adapt to a variety of partners and game types, whereas algorithms that rely on a single strategy or a less sophisticated set of experts are only successful in particular kinds of games played with particular partners[49] (Fig. 7c). Thus, in general, simplifying S# by removing experts from this set will tend to limit the algorithm's flexibility and generality, though doing so will not always negatively impact its performance when paired with particular partners in particular games.

Finally, the somewhat non-conventional expert-selection mechanism used by S# (see Eq. 1 in Methods) is central to its success. While techniques such as $\varepsilon$-greedy exploration (e.g., EEE[32]) and regret-matching (e.g., Exp3[29]) have permeated algorithm development in the AI community, S# instead uses an expert-selection mechanism closely aligned with recognition-primed decision-making[50]. Given the same full, rich set of experts, more traditional expert-selection mechanisms establish effective relationships in far fewer scenarios than S# (Fig. 7c). Figure 7 provides insights into why this is so. Compared to the other expert-selection mechanisms, S# has a greater combined ability to quickly establish a cooperative relationship with its partner (Fig. 7a) and then to maintain it (Fig. 7b), a condition

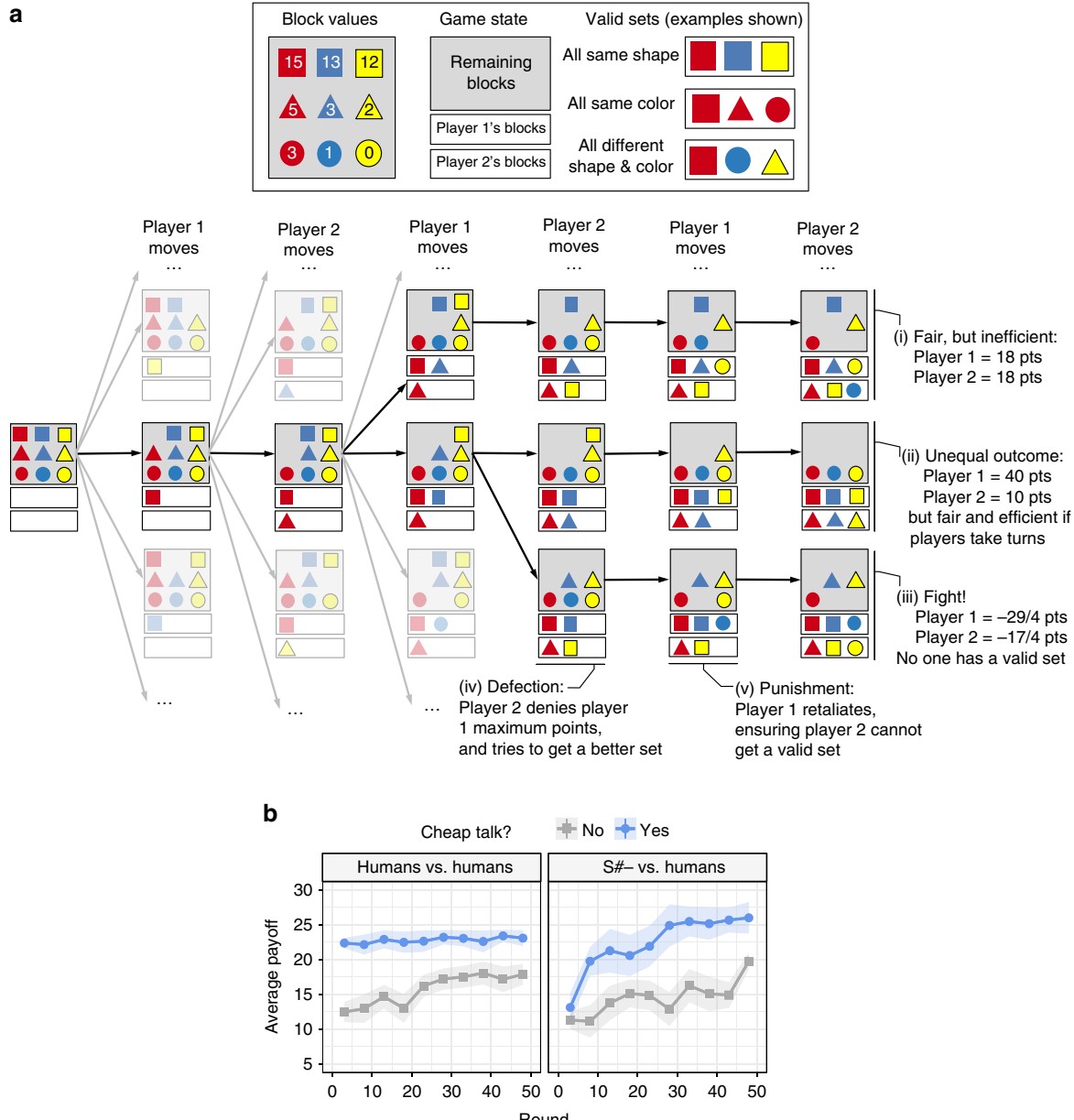

**Fig. 6** Results in a repeated stochastic game called the Block Game. **a** A partial view of a single round of the Block Game in which two players share a nine-piece block set. The two players take turns selecting blocks from the set until each has three blocks. The goal of each player is to get a valid set of blocks with the highest point value possible, where the value of a set is determined by the sum of the numbers on the blocks. Invalid sets receive negative points. The figure depicts five different potential outcomes for each round of the game, ranging from dysfunctional payoffs to outcomes in which one or both players benefit. The mutually cooperative (Nash bargaining) solution occurs when players take turns getting the highest quality set of three blocks (all the squares). **b** Average payoffs obtained by people and S#– (an early version of S# that generates, but does not respond to, cheap talk) when associating with people in the Block Game. Error ribbons show the standard error of the mean. As in normal-form games, S#– successfully uses cheap talk to consistently forge cooperative relationships with people in this repeated stochastic game. For more details, see Supplementary Note 6

brought about by S#'s tendency to not deviate from cooperation after mutual cooperation has been established (i.e., loyalty).

The loyalty brought about by S#'s expert-selection mechanism helps explain why S#–S# pairings substantially outperformed human–human pairings in our study (Fig. 4). S#'s superior performance can be attributed to two human tendencies. First, while S# did not typically deviate from cooperation after successive rounds of mutual cooperation (Fig. 7b), many human players did. Almost universally, such deviations led to reduced payoffs to the deviator. Second, as in human–human interactions observed in other studies[51], a sizable portion of our participants failed to keep some of their verbal commitments. On the other

hand, since S#'s verbal commitments are derived from its intended future behavior, it typically carries out the plans it proposes. Had participants followed S#'s strategy in these two regards (and all other behavior by the players had remained unchanged), human–human pairings could have performed nearly as well, on average, as S#–S# pairings (Fig. 8; see Supplementary Note 7 for details).

## Discussion

Our studies of human–S# partnerships were limited to five repeated games, selected carefully to represent different classes of

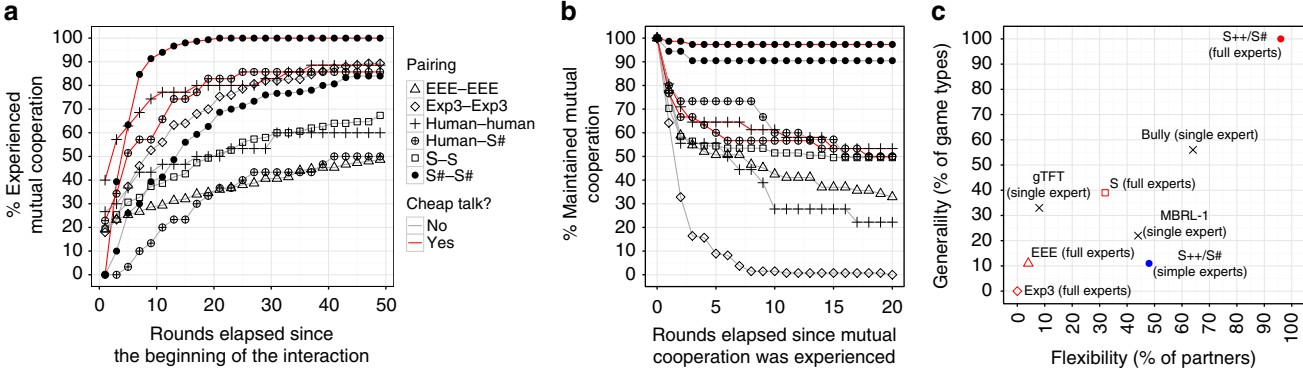

**Fig. 7** Comparisons of people and algorithms with respect to various characteristics. **a** Empirically generated cumulative-distribution functions describing the number of rounds required for pairings to experience two consecutive rounds of mutual cooperation across three games (Chicken, Alternator Game, and Prisoner's Dilemma). Per-game results are provided in Supplementary Note 7. For machine–machine pairings, the results are obtained from 50 trials conducted in each game, whereas pairings with humans use results from a total of 36 different pairings each. **b** The percentage of partnerships for each pairing that did not deviate from mutual cooperation once the players experienced two consecutive rounds of mutual cooperation across the same three repeated games. **c** A comparison of algorithms with respect to the ability to form profitable relationships across different games (Generality) and with different associates (Flexibility). Generality was computed as the percentage of game types (defined by payoff family × game length) for which an algorithm obtained the highest or second highest average payoffs compared to all 25 algorithms tested. Flexibility was computed as the percentage of associates against which an algorithm had the highest or second highest average payoff compared to all algorithms tested. See Supplementary Note 3 for details about each metric

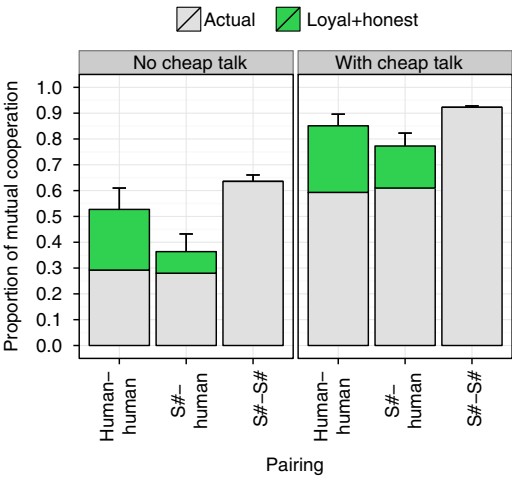

**Fig. 8** The estimated impact of honesty and loyalty. The estimated proportion of rounds that could have resulted in mutual cooperation had all human players followed S#'s learned behavioral and signaling strategies of not deviating from cooperative behavior when mutual cooperation was established (i.e., Loyal) and keeping verbal commitments (i.e., Honest), and all other behavior from the players remained unchanged. See Supplementary Note 7 for details of methods used. Error bars show the standard error of the mean

games from the periodic table of games (Supplementary Note 2). These games also included normal-form games as well as richer forms of RSGs. Though future work should address more scenarios, S#'s success in establishing cooperative relationships with people in these representative games, along with its consistently high performance across all classes of 2 × 2 games and various RSGs[44] when associating with other algorithms, gives us some confidence that these results will generalize to other scenarios.

This paper focused on the development and analysis of algorithmic mechanisms that allow learning algorithms to forge cooperative relationships with both people and other algorithms in two-player RSGs played with perfect information. This class of games encompasses a vast majority of cooperation problems

studied in psychology, economics, and political science. However, while the class of RSGs is quite general and challenging in and of itself, future work should focus on developing algorithms that can effectively cooperate with people and other algorithms in even more complex scenarios[52], including multi-player repeated games, repeated games with imperfect information, and scenarios in which the players possibly face a different payoff function in each round. We believe that principles and algorithmic mechanisms identified and developed in this work will help inform the development of algorithms that cooperate with people in these (even more challenging) scenarios.

Since Alan Turing envisioned AI, major milestones have often focused on either defeating humans in zero-sum encounters[1–6], or to interact with people as teammates that share a common goal[7–9]. However, in many scenarios, successful machines must cooperate with, rather than compete against, humans and other machines, even in the midst of conflicting interests and threats of being exploited. Our work demonstrates how autonomous machines can learn to establish cooperative relationships with people and other machines in repeated interactions. We showed that human–machine and machine–machine cooperation is achievable using a non-trivial, but ultimately simple, set of algorithmic mechanisms. These mechanisms include computing a variety of expert strategies optimized for various scenarios, a particular meta-strategy for selecting experts to follow, and the ability to generate and respond to costless, non-binding signals (called cheap talk) at levels conducive to human understanding. We hope that this extensive demonstration of human cooperation with autonomous machines in repeated games will spur significant further research that will ensure that autonomous machines, designed to carry out human endeavors, will cooperate with humanity.

## Methods

**Games for studying cooperation.** We describe the benchmark of games used in our studies, provide an overview of S++, and describe S# in more detail. Details that are informative but not essential to gaining an understanding of the main results of the paper are provided in the Supplementary Notes. We begin with a discussion about games for benchmarking cooperation.

We study cooperation between people and algorithms in long-term relationships (rather than one-shot settings[53]) in which the players do not share all

the same preferences. In this paper, we model these interactions as RSGs. An RSG, played by players $i$ and $-i$, consists of a set of rounds. In each round, the players engage in a sequence of stage games drawn from the set of stage games $S$. In each stage $s \in S$, both players choose an action from a finite set. Let $A(s) = A_i(s) \times A_{-i}(s)$ be the set of joint actions available in stage $s$, where $A_i(s)$ and $A_{-i}(S)$ are the action sets of players $i$ and $-i$, respectively. Each player simultaneously selects an action from its set of actions. Once joint action $\mathbf{a} = (a_i, a_{-i})$ is played in stage $s$, each player receives a finite reward, denoted $r_i(s, \mathbf{a})$ and $r_{-i}(s, \mathbf{a})$, respectively. The world also transitions to some new stage $s'$ with probability defined by $P_M(s, \mathbf{a}, s')$. Each round of an RSG begins in the start stage $\hat{s} \in S$ and terminates when some goal stage $s_g \in G \subseteq S$ is reached. A new round then begins in stage $\hat{s}$. The game repeats for an unknown number of rounds. In this work, we assume perfect information.

Real-world interactions often permit cheap talk, in which players send non-binding, costless signals to each other before taking actions. In this paper, we add cheap talk to repeated games by allowing each player, at the beginning of each round, to send a set of messages (selected from a pre-determined set of speech acts $M$) prior to acting in that round. Formally, let $\mathcal{P}(M)$ denote the power set of $M$, and let $m_i(t) \in \mathcal{P}(M)$ be the set of messages sent by player $i$ prior to round $t$. Only after sending the message $m_i(t)$ can player $i$ view the message $m_{-1}(t)$ (sent by its partner) and vice versa.

As with all historical grand challenges in AI, it is important to identify a class of benchmark problems to compare the performance of different algorithms. When it comes to human cooperation, a fundamental benchmark has been $2 \times 2$, general-sum, repeated games[54]. This class of games has been a workhorse for decades in the fields of behavioral economics[45], mathematical biology[45], psychology[46] sociology[47], computer science[36], and political science[26]. These fields have revealed many aspects of human cooperative behavior through canonical games, such as Prisoner's Dilemmas, Chicken, Battle of the Sexes, and the Stag Hunt. Such games, therefore, provide a well-established, extensively studied, and widely understood benchmark for studying the capabilities of machines to develop cooperative relationships.

Thus, for foundational purposes, we initially focus on two-player, two-action normal-form games, or RSGs with a single stage (i.e., $|S| = 1$). This allows us to fully enumerate the problem domain under the assumption that payoff functions follow strict ordinal preference orderings. The periodic table of $2 \times 2$ games[54–58] (see Supplementary Figure 1 along with Supplementary Note 2) identifies and categorizes 144 unique game structures that present many unique scenarios in which machines may need to cooperate. We use this set of game structures as a benchmark against which to compare the abilities of algorithms to cooperate. Successful algorithms should forge successful relationships with both people and machines across all of these repeated games. In particular, we can use these games to quantify the abilities of various state-of-the-art machine-learning algorithms to satisfy the properties advocated in the introduction: generality across games, flexibility across opponent types (including humans), and speed of learning.

Though we initially focus on normal-form RSGs, we are interested in algorithms that can be used in more general settings, such as RSGs in which $|S| > 1$. These games require players to reason over multiple actions in each round. Thus, we also study the algorithms in a set of such games, including the Block Game shown in Fig. 6a. Additional results are reported in Supplementary Note 6.

The search for metrics that properly evaluate successful behavior in repeated games has a long history, for which we refer the reader to Supplementary Note 2. In this paper, we focus on two metrics of success: empirical performance and proportion of mutual cooperation. Ultimately, the success of a player in an RSG is measured by the sum of the payoffs the player receives over the duration of the game. A successful algorithm should have high empirical performance across a broad range of games when paired with many different kinds of partners. However, since the level of mutual cooperation (i.e., how often both players cooperate with each other) often highly correlates with a player's empirical performance[26], the ability to establish cooperative relationships is a key attribute of successful algorithms. However, we do not consider mutual cooperation as a substitute for high empirical performance, but rather as a supporting factor.

The term "cooperation" has specific meaning in well-known games such as the Prisoner's Dilemma. In other games, the term is much more nebulous. Furthermore, mutual cooperation can be achieved in degrees; it is usually not an all or nothing event. However, for simplicity in this work, we define mutual cooperation as the Nash bargaining solution of the game[59], defined as the unique solution that maximizes the product of the players' payoffs minus their maximin values. Supplementary Table 4 specifies the Nash bargaining solutions for the games used in our user studies. Interestingly, the proportion of rounds that players played mutually cooperative solutions (as defined by this measure) was strongly correlated with the payoffs a player received in our user studies. For example, in our third user study, the correlation between payoffs received and proportion of mutual cooperation was $r(572) = 0.909$.

## Overview of S++.
S# is derived from S++[37,44], an expert algorithm that combines and builds on decades of research in computer science, economics, and the behavioral and social sciences. Since understanding S++ is key to understanding S#, we first overview S++.

S++ is defined by a method for computing a set of experts for arbitrary RSGs and a method for choosing which expert to follow in each round (called the expert-selection mechanism). Given a set of experts, S++'s expert-selection mechanism uses a meta-level control strategy based on aspiration learning[34,60,61] to

dynamically prune the set of experts it considers following in a round. Formally, let $E$ denote the set of experts computed by S++. In each epoch (beginning in round $t$), S++ computes the potential $\rho_j(t)$ of each expert $e_j \in E$, and compares this potential with its aspiration level $\alpha(t)$ to form the reduced set $E(t)$ of experts:

$$E(t) = \{e_j \in E : \rho_j(t) \geq \alpha(t)\}. \qquad (1)$$

This reduced set consists of the experts that S++ believes could potentially produce satisfactory payoffs. It then selects one expert $e_{\text{sel}}(t) \in E(t)$ using a satisficing decision rule[34,60]. Over the next $m$ rounds, it follows the strategy prescribed by $e_{\text{sel}}(t)$. After these $m$ rounds, it updates its aspiration level as follows:

$$\alpha(t + m) \leftarrow \lambda^m \alpha(t) + (1 - \lambda^m) R, \qquad (2)$$

where $\lambda \in (0, 1)$ is the learning rate and $R$ is the average payoff S++ obtained in the last $m$ rounds. It also updates each expert $e_j \in E$ based on its peculiar reasoning mechanism. A new epoch then begins.

S++ uses the description of the game environment to compute a diverse set of experts. Each expert uses distinct mathematics and assumptions to produce a strategy over the entire space of the game. The set of experts used in the implementation of S++ used in our user studies includes five expectant followers, five trigger strategies, a preventative strategy[44], the maximin strategy, and a model-based reinforcement learner (MBRL-1). For illustrative purposes relevant to the description of S#, we overview how the expectant followers and trigger strategies are computed.

Both trigger strategies and expectant followers (which are identical to trigger strategies except that they omit the punishment phase of the strategy) are defined by a joint strategy computed over all stages of the RSG. Thus, to create a set of such strategies, S++ first computes a set of pareto-optimal joint strategies, each of which offers a different compromise between the players. This is done by solving Markov decision processes (MDPs) over the joint-action space of the RSG. These MDPs are defined by $A$, $S$, and $P_M$ of the RSG, as well as a payoff function defined as a convex combination of the players' payoffs[62]:

$$y^\omega(s, \mathbf{a}) = \omega r_i(s, \mathbf{a}) + (1 - \omega) r_{-i}(s, \mathbf{a}), \qquad (3)$$

where $\omega \in [0, 1]$. Then, the value of joint-action $\mathbf{a}$ in state $s$ is

$$Q^\omega(s, \mathbf{a}) = y^\omega(s, \mathbf{a}) + \sum_{s' \in S} P_M(s, \mathbf{a}, s') V^\omega(s'), \qquad (4)$$

where $V^\omega(s) = \max_{\mathbf{a} \in A(s)} Q^\omega(s, \mathbf{a})$. The MDP can be solved in polynomial time using linear programming[63].

By solving MDPs of this form for multiple values of $\omega$, S++ computes a variety of possible pareto-optimal[62] joint strategies. We call the resulting solutions "pure solutions." These joint strategies produce joint payoff profiles. Let MDP($\omega$) denote the joint strategy produced by solving an MDP for a particular $\omega$. Also, let $V_i^\omega(s)$ be player $i$'s expected future payoff from stage $s$ when MDP($\omega$) is followed. Then, the ordered pair $\left(V_i^\omega(\hat{s}), V_{-i}^\omega(\hat{s})\right)$ is the joint payoff vector for the strategy defined by MDP($\omega$). Additional solutions, or "alternating solutions," are obtained by alternating across rounds between different pure solutions. Since longer cycles are difficult for a partner to model, S++ only includes cycles of length two.

Applying this technique to a 0-1-3-5 Prisoner's Dilemma produces the joint payoffs depicted in Fig. 9a. Notably, MDP(0.1), MDP(0.5), and MDP(0.9) produce the joint payoffs (0, 5), (3, 3), and (5, 0), respectively. Alternating between these solutions produces three other solutions whose (average) payoff profiles are also shown. These joint payoffs reflect different compromises, of varying degrees of fairness, that the two players could possibly agree upon.

Regardless of the structure of the RSG, including whether it is simple or complex, this technique produces a set of potential compromises available in the game. For example, Fig. 9b shows potential solutions computed for a two-player micro-grid scenario[44] (with asymmetric payoffs) in which players must share energy resources. Despite the differences in the dynamics of the Prisoner's Dilemma and this micro-grid scenario, these games have similar sets of potential compromises. As such, in each game, S++ must learn which of these compromises to play, including whether to make fair compromises, or compromises that benefit one player over the other (when they exist). These game-independent similarities can be exploited by S# to provide signaling capabilities that can be used in arbitrary RSG's, an observation we exploited to develop S#'s signaling mechanisms (see the next subsection for details).

The implementation of S++ used in our user studies selects five of these compromises, selected to reflect a range of different compromises. They include the solution most resembling the game's egalitarian solution[62], and the two solutions that maximize each player's payoff subject to the other player receiving at least its maximin value (if such solutions exist). The other two solutions are selected to maximize the Euclidean distance between the payoff profiles of selected solutions. The five selected solutions form five expectant followers (which simply repeatedly follow the computed strategy) and five trigger strategies. For the trigger strategies, the selected solutions constitute the offers. The punishment phase is the strategy that minimizes the partner's maximum expected payoff, which is played after the partner deviates from the offer until the sum of its partner's payoffs (from the time of the

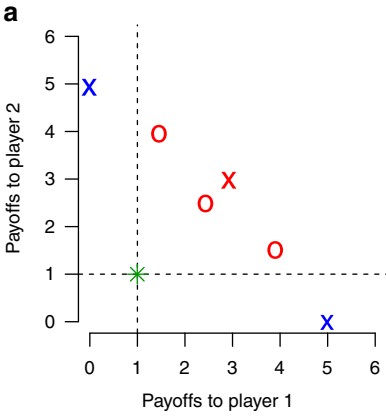

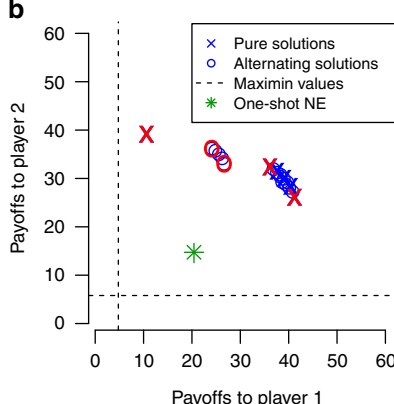

**Fig. 9** Target solutions computed by S++. The x's and o's are the joint payoffs of possible target solutions computed by S++. Though games vary widely, the possible target solutions computed by S++ in each game represent a range of different compromises, of varying degrees of fairness, that the two players could possibly agree upon. For example, tradeoffs in solution quality are similar for **a** a 0-1-3-5 Prisoner's Dilemma and **b** a more complex micro-grid scenario[44]. Solutions marked in red are selected by S++ as target solutions

deviation) are below what it would have obtained had it not deviated from the offer. This makes following the offer of the trigger strategy the partner's optimal strategy.

The resulting set of experts available to S++ generalizes many popular strategies that have been studied in past work. For example, for prisoner's dilemmas, the computed trigger strategies include Generous Tit-for-Tat and other strategies that resemble zero-determinant strategies[64] (e.g., Bully[65]). Furthermore, computed expectant followers include Always Cooperate, while the expert MBRL-1 quickly learns the strategy Always Defect when paired with a partner that always cooperates. In short, the set of strategies (generalized to the game being played) available for S++ to follow give it the flexibility to effectively adapt to many different kinds of partners.

**Description of S#.** S# builds on S++ in two ways. First, it generates speech acts to try to influence its partner's behavior. Second, it uses its partner's speech acts to make more informed choices regarding which experts to follow.

Signaling to its partner: S#'s signaling mechanism was designed with three properties in mind: game independence, fidelity between signals and actions, and human understandability (i.e., the signals should be communicated at a level conducive to human understanding). One way to do this is to base signal generation on game-independent, high-level ideals rather than game-specific attributes. Example ideals include proficiency assessment[10,27,61,66], fairness[67,68], behavioral expectations[25], and punishment and forgiveness[26,69]. These ideals package well-established concepts of interaction in terms people are familiar with.

Not coincidentally, S++'s internal state and algorithmic mechanisms are largely defined in terms of these high-level ideals. First, S++'s decision-making is governed by proficiency assessment. It continually evaluating its own proficiency and that of its experts by comparing its aspiration level (Eq. 2), which encodes performance expectations, with its actual and potential performance (see descriptions of traditional aspiration learning[34,60,61] and Eq. 1). S# also evaluates its partner's performance against the performance it expects it partner to have. Second, the array of compromises computed for expectant followers and trigger strategies encode various degrees of fairness. As such, these experts can be defined and even referred to by references to fairness. Third, strategies encoded by expectant followers and trigger strategies define expectations for how the agent and its partner should behave. Finally, transitions between the offer and punishment phases of trigger strategies define punishment and forgiveness.

The cheap talk generated by S# is based not on the individual attributes of the game, but rather on events taken in context of these five game-independent principles (as they are encoded in S++). Specifically, S# automatically computes a finite-state machine (FSM) with output (specifically, a Mealy machine[70]) for each expert. The states and transitions in the state machine are defined by proficiency assessments, behavioral expectations, and (in the case of experts encoding trigger strategies) punishment and forgiveness. The outputs of each FSM are speech acts that correspond to the various events and that also refer to the fairness of outcomes and potential outcomes.

For example, consider the FSM with output for a trigger strategy that offers a pure solution, which is shown in Fig. 3b. States $s_0 - s_6$ are states in which S# has expectations that its partner will conform with the trigger strategy's offer. Initially (when transitioning from state $s_0$ to $s_1$), S# voices these behavioral expectations (speech act #15), along with a threat that if these expectations are not met, it will punish its partner (speech act #0). If these expectations are met (event labeled $s$), S# praises its partner (speech acts #5–6). On the other hand, when behavioral expectations are not met (events labeled $d$ and $g$), S# voices its dissatisfaction with its partner (speech acts #11–12). If S# determines that its partner has benefitted from the deviation (proficiency assessment), the expert transitions to state $s_7$, while telling its partner that it will punish him (speech act #13). S# stays in this

punishment phase and voices pleasure in reducing its partner's payoffs (speech act #14) until the punishment phase is complete. It then transitions out of the punishment phase (into state $s_8$) and expresses that it forgives its partner, and then returns to states in which it renews behavioral expectations for its partner.

FSMs for other kinds of experts along with specific details for generating them, are given in Supplementary Note 4.

Because S#'s speech generation is based on game-independent principles, the FSMs for speech generation are the same for complex RSGs as they are for simple (normal-form) RSGs. The exception to this statement is the expression of behavioral expectations, which are expressed in normal-form games simply as sequences of joint actions. However, more complex RSGs (such as the Block Game; Fig. 6) have more complex joint strategies that are not as easily expressed in a generic way that people understand. In these cases, S# uses game-invariant descriptions of fairness to specify solutions, and then depends on its partner to infer the details. It refers to a solution in which players get similar payoffs as a "cooperative" or "fair" solution, and a solution in which one player scores higher than the other as a solution in which "you (or I) get the higher payoff." While not as specific, our results demonstrate that such expressions can be sufficient to communicate behavioral expectations.

While signaling via cheap talk has great potential to increase cooperation, honestly signaling one's internal states exposes a player to the potential of being exploited. Furthermore, the so-called silent treatment is often used by humans as a means of punishment and an expression of displeasure. For these two reasons, S# also chooses not to speak when its proposals are repeatedly not followed by its partner. The method describing how S# determines whether or not to voice speech acts is described in Supplementary Note 4.

In short, S# essentially voices the stream of consciousness of its internal decision-making mechanisms, which are tied to the aforementioned game-independent principles. Since these principles also tend to be understandable to humans and are present in all forms of RSGs, S#'s signal-generation mechanism tends to achieve the three properties we desired to satisfy: game independence, fidelity between signals and actions, and human understandability.

Responding to its partner: In addition to voicing cheap talk, the ability to respond to a partner's signals can substantially enhance one's ability to quickly coordinating on cooperative solutions (see Supplementary Note 4, including Supplementary Table 12). When its partner signals a desire to play a particular solution, S# uses proficiency assessment to determine whether it should consider playing it. If this assessment indicates that the proposed solution could be a desirable outcome to S#, it determines which of its experts play strategies consistent (or congruent) with the proposed solution to further reduce the set of experts that it considers following in that round (step 3 in Fig. 3a). Formally, let $E_{cong}(t)$ denote the set of experts in round $t$ that are congruent with the last joint plan proposed by S#'s partner. Then, S# considers selecting experts from the set defined as:

$$E(t) = \{e_j \in E_{cong}(t) : \rho_j(t) \geq \alpha(t)\}. \quad (5)$$

If this set is empty (i.e., no desirable options are congruent with the partner's proposal), $E(t)$ is calculated as with S++ (Eq. 1).

The congruence of the partner's proposed plan with an expert is determined by comparing the strategy proposed by the partner with the solution espoused by the expert. In the case of trigger strategies and expectant followers, we compare the strategy proposed by the partner with the strategies' offer. This is done rather easily in normal-form games, as the partner can easily and naturally express its strategy as a sequence of joint actions, which is then compared to the sequence of joint actions of the expert's offer. For general RSGs, however, this is more difficult because a

joint action is viewed as a sequence of joint strategies over many stages. In this case, S# must rely on high-level descriptions of the solution. For example, solutions described as "fair" and "cooperative" can be assumed to be congruent with solutions that have payoff profiles similar to those of the Nash bargaining solution.

Listening to its partner can potentially expose S# to being exploited. For example, in a (0-1-3-5)-Prisoner's Dilemma, a partner could continually propose that both players cooperate, a proposal S# would continually accept and act on when $\alpha_t^i \leq 3$. However, if the partner did not follow through with its proposal, it could potentially exploit S# to some degree for a period of time. To avoid this, S# listens to its partner less frequently the more the partner fails to follow through with its own proposals. The method describing how S# determines whether or not to listen to its partner is described in Supplementary Note 4.

**User studies and statistical analysis**. Three user studies were conducted as part of this research. Complete details about these studies and the statistical analysis used to analyze the results are given in Supplementary Notes 5–7.

**Data availability**. The data sets from our user studies, the computer code used to generate the comparison of algorithms, and our implementation of S# can be obtained by contacting Jacob Crandall.

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

## Acknowledgements

We would like to acknowledge Vahan Babushkin and Sohan D'Souza for their help in conducting user studies. J.W.C. was supported by donations made to Brigham Young University and by Masdar Institute. M.O., T., and F.I.-O. were also supported by Masdar Institute. J.-F.B. gratefully acknowledges support through the ANR-Labex IAST. I.R. is supported by the Ethics and Governance of Artificial Intelligence Fund.

## Author contributions

J.W.C. was primarily responsible for algorithm development, with Tennom and M.O. also contributing. J.W.C., M.O., Tennom, F.I.-O., and I.R. designed and conducted user studies. J.W.C. and S.A. wrote code for the comparison of existing algorithms. J.-F.B. led the statistical analysis, with M.O., F.I.-O., J.W.C., and I.R. also contributing. I.R., J.W.C., M.C., J.-F.B., A.S., and S.A. contributed to the interpretation of the results and framing of the paper. All authors, with J.W.C. and I.R. leading, contributed to the writing of the paper. The vision of M.A.G. started it all many years ago.

## Additional information

**Competing interests:** The authors declare no competing financial interests.

