## [Peer Review File · Nature Communications]

Reviewers' comments:

Reviewer #1 (Remarks to the Author):

The paper investigates the question how machines can cooperate with each other and with humans.

More specifically:

The paper provides a detailed empirical comparison of 25 algorithms from the literature wrt to their ability to cooperate in simple normal form games and repeated stochastic games. The selection of the algorithms is comprehensive and has representatives from all relevant families of algorithms that make sense. The set of games/tasks considered for the benchmark is chosen well, including an exhaustive set of 2x2 normal form games, as well as interesting RSGs. The authors evaluate the algorithms under different number of interactions as well.

The paper proposes a new algorithm S#, as an extension of the S++ algorithm (which comes out as the winner of the benchmark). The S# algorithm extends S++ in that it allows for the incorporation of "cheap talk" into the interaction, both by executing speech acts as well as taking into account observed speech acts by the interaction partner.

Finally, the paper evaluates the new S# algorithm in the context of the proposed benchmarks with "cheap talk" allowed as well. This evaluation also includes human interaction partners. The evaluation shows that the S# algorithm is able to make use of "cheap talk" and achieves impressive results: Both with and without cheap talk, it achieves results with a human partner that are very similar to a human-human interaction. In addition, when interacting with itself, it achieves even better results in terms of total payoff and number of cooperative interactions.

All three sets of results presented in this paper are original.

The benchmark study is unique and there is no study of comparable thoroughness and exhaustiveness. The results will be of great interest to the community and are likely to initiate further research in this area.

The algorithm S# appears to be original not only in its algorithmic details, but also in its purpose. I am not aware of previous algorithms that were able to interact with other agents using cheap talk to promote their goals. As a new domain and problem setting, this will be of great interest to the AI community as well as in psychology, economics and related fields.

The evaluation of the S# algorithm with other algorithms and humans is also original and interesting. In particular, the aspect that the algorithm is able to communicate with humans both as a sender and receiver of cheap talk is of great interest, even more so, as the algorithm outperforms humans in self-play, even if those humans are allowed cheap talk as well.

I believe that this is a ground-breaking paper. Especially now that the role of AI, and its interaction with humans becomes an ever more interesting and pressing question, this paper provides not only a very interesting set of benchmark problems for evaluation, but it proposes an interesting class of algorithms that may be able to establish and maintain cooperation in mixed incentive games. In addition, the paper highlights the role of “cheap talk” for building and maintaining cooperation, a notion that had until this point mostly been associated with human approaches to these problems. Successfully employing cheap talk in computer-computer interactions as well as computer-human interactions is astounding! While the algorithm to achieve this is a little too complicated by my standards, it represents a great first step and is sure to stimulate more interesting research in this important and fascinating area.

While the overall goals and results of the paper are convincing, parts of the analysis need improvement.

An open question is, in how far the success of the S# algorithm with cheap talk is due to sending or receiving cheap talk. I would recommend to amend the study with an ablation study, where the algorithm is tested with a) only sending cheap talk, b) only receiving cheap talk, and c) both sending and receiving cheap talk.

Regarding the statistical analysis, I would request a more hypothesis driven procedure. For example in Figure 3, there are bar charts about Speech Profiles, Subjective Evaluations, and Turing Test. However, many of the bars are similar in height. Please pose statistical hypotheses and provide test statistics. For example, does the data refute the hypothesis that “Thought to be human” is the same for both cases with cheap talk and without cheap talk?

I did not understand Figure 5 c). Please explain flexibility, generality, and how they are used and interpreted here.

I did not understand the counterfactual argument made regarding Figure 6. How can the authors possibly state what would have happened in human-human pairings had they behaved differently? I understand, how this might be an interesting question, but I doubt that there is a principled methodology for finding an answer here. This would require a model of human-human play that would be able to answer counterfactual questions.

Section 3.1.: What do you mean by the last sentence regarding maximising the product of the players advantages? What is an advantage (technical term?) Is this equivalent to the NBS?

Section 3.2.: Last paragraph. How is cooperation defined for arbitrary games? How can one determine 30% cooperation on that basis?

In summary, I recommend accepting the paper once the above points have been addressed and the statistical analysis has been made rigorous in a hypothesis/test driven way.

Reviewer #2 (Remarks to the Author):

I'm not an expert on AI or its application to experimental games and therefore don't have the background needed to comment on parts of the paper, especially the technical ones. As the authors acknowledge, experimental games are idealized models of real life social interaction, and their algorithms are tested within the narrow confines of that model. Nevertheless, I read the paper with great interest and was impressed with the evidence for the S# algorithm's ability to be as effective as humans in collaborating with other humans and with the evidence tracing the performance of the algorithm to the use of verbal communication, the consistent cooperation after mutual cooperation is achieved, and to the adherence to verbal commitment.

As a psychologist, two things caught my attention:

First, although many participants didn't pass the Turing test, Figure 3(c) shows that, overall, they were more likely to identify that S# rather than a human is a machine. This raises the question whether participants' identification of their partner as a machine vs. human predicts performance. It is possible that participants have less interest in competing with a machine than with another human and, in fact, are less competitive when they think their partner is just a machine. I'm concerned that this could have contributed to the finding that the collaboration of participants with S# matches their collaboration with other human participants.

Second, Figure 3(a) shows that S#, compared to humans, used more affectively charged negative messages--more hateful, threatening, and less praiseful. Does the use of such messages predict performance? Does this depend on whether the partner is a machine (or thought to be a machine) rather than a human being? And if there is such dependency, does it explain the finding that the collaboration of participants with S# matches their collaboration with other human participants?

Overall, I learned a lot from this paper and think that a revision that addresses these would be a good candidate for publication in this journal.

Reviewer #3 (Remarks to the Author):

The paper considers a very interesting and important problem.

However, at its current state it could not be accepted to Nature Communications.

The paper is not well written. Its main contribution is a modified algorithm for S++, called S#, aimed at generating non-binding, costless communication with people for establishing relationships.

S++ is only described on page 13, and S# on page 16. It is hard to understand anything that is presented before that as many terms are used without proper definitions or examples. Surprisingly, the results are given without the full experimental details which were omitted "in the interest of space". The results seem significant but it's hard to understand exactly what is the significance of this paper as the modification of S++ is marginal and the experimental results are reported for only five, very simple repeated game settings. I did like the extensive comparison with other algorithms.

Reviewer #1 (Remarks to the Author):

The paper investigates the question how machines can cooperate with each other and with humans.

More specifically:

The paper provides a detailed empirical comparison of 25 algorithms from the literature wrt to their ability to cooperate in simple normal form games and repeated stochastic games. The selection of the algorithms is comprehensive and has representatives from all relevant families of algorithms that make sense. The set of games/tasks considered for the benchmark is chosen well, including an exhaustive set of 2x2 normal form games, as well as interesting RSGs. The authors evaluate the algorithms under different number of interactions as well. The paper proposes a new algorithm S#, as an extension of the S++ algorithm (which comes out as the winner of the benchmark). The S# algorithm extends S++ in that it allows for the incorporation of “cheap talk” into the interaction, both by executing speech acts as well as taking into account observed speech acts by the interaction partner.

Author response: We welcome the reviewer’s positive feedback on the scale and representativeness of the comparison, since identifying algorithms that reflect the state-of-the-art in general-sum games is important to the paper’s contributions.

Finally, the paper evaluates the new S# algorithm in the context of the proposed benchmarks with “cheap talk” allowed as well. This evaluation also includes human interaction partners. The evaluation shows that the S# algorithm is able to make use of “cheap talk” and achieves impressive results: Both with and without cheap talk, it achieves results with a human partner that are very similar to a human-human interaction. In addition, when interacting with itself, it achieves even better results in terms of total payoff and number of cooperative interactions.

All three sets of results presented in this paper are original.

The benchmark study is unique and there is no study of comparable thoroughness and exhaustiveness. The results will be of great interest to the community and are likely to initiate further research in this area.

The algorithm S# appears to be original not only in its algorithmic details, but also in its purpose. I am not aware of previous algorithms that were able to interact with other agents using cheap

talk to promote their goals. As a new domain and problem setting, this will be of great interest to the AI community as well as in psychology, economics and related fields.

The evaluation of the S# algorithm with other algorithms and humans is also original and interesting. In particular, the aspect that the algorithm is able to communicate with humans both as a sender and receiver of cheap talk is of great interest, even more so, as the algorithm outperforms humans in self-play, even if those humans are allowed cheap talk as well.

I believe that this is a ground-breaking paper. Especially now that the role of AI, and its interaction with humans becomes an ever more interesting and pressing question, this paper provides not only a very interesting set of benchmark problems for evaluation, but it proposes an interesting class of algorithms that may be able to establish and maintain cooperation in mixed incentive games. In addition, the paper highlights the role of “cheap talk” for building and maintaining cooperation, a notion that had until this point mostly been associated with human approaches to these problems. Successfully employing cheap talk in computer-computer interactions as well as computer-human interactions is astounding! While the algorithm to achieve this is a little too complicated by my standards, it represents a great first step and is sure to stimulate more interesting research in this important and fascinating area.

Author response: We are grateful to the reviewer for these encouraging remarks.

While the overall goals and results of the paper are convincing, parts of the analysis need improvement.

An open question is, in how far the success of the S# algorithm with cheap talk is due to sending or receiving cheap talk. I would recommend to amend the study with an ablation study, where the algorithm is tested with a) only sending cheap talk, b) only receiving cheap talk, and c) both sending and receiving cheap talk.

Author response: The reviewer raises an interesting question which we had considered, but did not adequately articulate and fully evaluate previously. Indeed, both generating (talking) and responding to (listening) cheap talk play important roles in S#'s effectiveness. We have rewritten Section C.3 in the supplementary information in response to this recommendation. This section now includes a description of a new evaluation of the impact of both talking and listening on S#'s ability to forge cooperative relationships. We now reference this additional study on pages 3 and 18 of the revised manuscript to highlight the importance of both generating and responding to cheap talk.

Initially, we created S#-, an algorithm that generated cheap talk (i.e., it talked), but did not respond to its partner's messages (i.e., it did not listen). In effect, this meant that information was passed via cheap talk in a single direction (from S# to the human participant), but not in the other direction (from the human to S#). This preliminary version of S# was used in User Study 2 (reported in detail in SI.E), and we now have modified Section 1.3 of the main manuscript to better emphasize this. While generating cheap talk increased the algorithm's ability to forge

cooperative (and more profitable) relationships with people (above that of S++, which does not use cheap talk), this one-way communication did not produce as consistent and fast convergence to cooperative solutions as we desired. Furthermore, we found that people become somewhat frustrated while they waited for the machine to propose an adequate solution they could accept. Thus, we added two-way communication in User Study 3 (reported in detail in SI.F).

In the revised manuscript, we went one step further to formalize the impact of both listening and talking (as opposed to just doing one or the other). We conducted an additional evaluation, and described this additional evaluation in the revised Section C.3. This ablation study compares how (1) both listening and talking (communication via cheap between the players talk flows two ways) impact cooperation compared to (2) not listening or talking (No communication) or (3) just listening or just talking (one-way communication) when the agent is paired with S# (note that “just listening” or “just talking” ends up being the same in this evaluation -- information is understood in only a single direction regardless). In each of the games we considered in this evaluation, just listening or just talking is better than not communicating via cheap talk at all, but not as effective as both listening and talking. For the reviewer’s convenience, the figure depicting the results is given below -- full details, along with statistical analysis, are provided in the SI.

Regarding the statistical analysis, I would request a more hypothesis driven procedure. For example in Figure 3, there are bar charts about Speech Profiles, Subjective Evaluations, and Turing Test. However, many of the bars are similar in height. Please pose statistical hypotheses and provide test statistics. For example, does the data refute the hypothesis that “Thought to be human” is the same for both cases with cheap talk and without cheap talk?

Author response: In the revised manuscript, we now report statistical analyses of all the results displayed in Figure 3. Regarding speech profiles, S# used significantly more Hate and Threats messages, and significantly less Praise messages than humans. There was no statistically significant differences for Manage and Planning messages. Regarding ratings of messages,

there was no statistically significant differences in how users rated the intelligence, clarity of intent, and usefulness of the messages when messages are sent by S# vs. other humans. Finally, our analysis showed no statistically significant difference between “thought to be human” based on whether the partner is human vs. S#. Cheap talk had a marginally statistically significant effect. We did not detect any interaction between these two predictors.

We added overviews of these statistical results in the caption of Figure 3, and added the full analyses in Section F.3.3 of the SI.

I did not understand Figure 5 c). Please explain flexibility, generality, and how they are used and interpreted here.

Author response: It was an oversight on our part to not define these terms in the figure caption and we are glad to have an opportunity to correct it in the revised manuscript. We have corrected this in the figure caption. We also added details to Section B.5.3 in the SI to remove ambiguity.

In short, as stated in the revised figure 5 caption, “*Generality* was computed as the percentage of games types (defined by payoff family x game length) for which an algorithm obtained the highest or second highest average payoffs compared to all 25 algorithms tested. *Flexibility* was computed as the percentage of associates against which an algorithm had the highest or second highest average payoff compared to all algorithms tested. See SI.B.5.3 for details about each metric.”

I did not understand the counterfactual argument made regarding Figure 6. How can the authors possibly state what would have happened in human-human pairings had they behaved differently? I understand, how this might be an interesting question, but I doubt that there is a principled methodology for finding an answer here. This would require a model of human-human play that would be able to answer counterfactual questions.

Author response: We intend these results to be more of an investigation of what the players could have obtained had they all been honest and loyal (as S# typically was), rather than an assessment of what would definitely have happened counterfactually. In other words, this analysis quantifies loss of welfare, rather than making claims about counterfactual causality. We have tried to reword the sentence that references Figure 6 in the main text, the Figure 6 caption, and the results in the supplementary material to reflect this.

We agree that some clarification is in order here in the main manuscript. Figure 6 quantifies the loss of welfare compared to what would have happened had:

- (1) Both players always been honest
- (2) Both players always been loyal
- (3) All other behavior by the players remained unchanged

We did two things to derive Figure 6:

1. We looked at times in which a player proposed that the players play the mutually cooperative solution of the game and did not follow that solution, but their partner did follow the proposed solution. These are instances in which a player is considered to be dishonest. Had the partner been honest (and the other player's behavior had still done the same thing as we actually observed them do), the players would have begun playing the mutually cooperative solution. Thus, if both players were also loyal thereafter, we know that the mutually cooperative solution would have been played for the remaining rounds of the game.
2. We look at times in which the mutually cooperative solution had been played for two consecutive previous rounds, but then one or more of the players decided to not cooperate in the next round (disloyal behavior). This disloyal behavior often led to many subsequent rounds of "defection" by one or more players. We observe that if both players had been loyal (meaning neither of them would have broken the pattern of mutual cooperation), the mutually cooperative solution would have been played for the remaining rounds of the game.

Thus, the three previously mentioned assumptions define sequences of rounds of the game that the mutually cooperative solution would have been played. We then compare this outcome with what actually happened to derive Figure 6. We also analyze in the SI (SI.F.4.3) that both players almost always had lower individual payoffs (over the course of a game) when they failed to be honest and loyal compared to what they would have received had both players cooperated.

In the revised manuscript, we attempted to better explain the analysis in Figure 6 by doing two things:

1. We altered the last sentence of Section 1 to (1) emphasize all of the previously mentioned assumptions and to (2) provide a pointer where the methods used to derive the figure are described in the SI. This sentence now reads as follows: "Had participants followed S#'s strategy in these two regards (and all other behavior by the players had remained unchanged), Human-Human pairings could have performed nearly as well, on average, as S#-S# pairings (Figure 6; see SI.F.4.3 for details)."
2. We altered the caption of Figure 3 to explicitly state the three assumptions made in this analysis, as we had failed in the previous version to mention the third assumption in the caption. The new caption reads as follows: "The estimated proportion of rounds that could have resulted in mutual cooperation had (1) all human players followed S#'s learned behavioral and signaling strategies of not deviating from cooperative behavior when mutual cooperation was established (i.e., Loyal) and keeping verbal commitments (i.e., Honest), and (2) all other behavior from the players remained unchanged. See SI.F.4.3 for details of methods used. Error bars show the standard error of the mean."

Section 3.1.: What do you mean by the last sentence regarding maximising the product of the players advantages? What is an advantage (technical term?) Is this equivalent to the NBS?

Author response: This sentence was indeed vague. In order to clarify what we meant, we have revised the last paragraph of Section 3.1 to read as follows in the revised manuscript: “The term *cooperation* has specific meaning in well-known games such as the Prisoner's Dilemma. In other games, the term is much more nebulous. Furthermore, mutual cooperation can be achieved in degrees; it is usually not an all or nothing event. However, for simplicity in this work, we define *mutual cooperation* as the *Nash bargaining solution* of the game [57], defined as the unique solution that maximizes the product of the players' payoffs minus their maximin values. Table 4 in the SI (page 14) specifies the Nash bargaining solutions for the games used in our user studies. Interestingly, the proportion of rounds in which players played mutually cooperative solutions (as defined by this measure) was strongly correlated with the payoffs a player received in our user studies (e.g., in the third user study, $r(572)=0.909, p < 0.001$).

Section 3.2.: Last paragraph. How is cooperation defined for arbitrary games? How can one determine 30% cooperation on that basis?

Author response: As discussed at the end of Section 3.1, we have defined and measured mutual cooperation as the Nash bargaining solution of the game. Thus, 30% cooperation refers to the (approximate) percentage of rounds in which the players played the Nash bargaining solutions. Table 4 in the supplementary material (page 14) shows these solutions for the games considered in our user studies. In the revised manuscript, we now reference this table to give additional clarity to the reader, so that they can more easily interpret the Nash bargaining solution. See also our response to the previous question.

In summary, I recommend accepting the paper once the above points have been addressed and the statistical analysis has been made rigorous in a hypothesis/test driven way.

In addition to providing the additionally requested statistical analysis, we have also added pointers to the full statistical analysis provided in the SI in the figure captions. We believe that we have now conducted and reported a thorough statistical analysis of all the results.

Reviewer #2 (Remarks to the Author):

I'm not an expert on AI or its application to experimental games and therefore don't have the background needed to comment on parts of the paper, especially the technical ones. As the authors acknowledge, experimental games are idealized models of real life social interaction, and their algorithms are tested within the narrow confines of that model. Nevertheless, I read the paper with great interest and was impressed with the evidence for the S# algorithm's ability to be as effective as humans in collaborating with other humans and with the evidence tracing the performance of the algorithm to the use of verbal communication, the consistent cooperation after mutual cooperation is achieved, and to the adherence to verbal commitment.

Author response: We appreciate the reviewer's openness to the work. –The paper was only made possible through a collaboration among authors from different disciplines; and since we submitted this manuscript to a multi-disciplinary journal, it is important for us to incorporate

feedback from experts in a variety of fields (not just AI). So we appreciate the reviewer's comments, which helped us improve the paper.

As a psychologist, two things caught my attention:

First, although many participants didn't pass the Turing test, Figure 3(c) shows that, overall, they were more likely to identify that S# rather than a human is a machine. This raises the question whether participants' identification of their partner as a machine vs. human predicts performance. It is possible that participants have less interest in competing with a machine than with another human and, in fact, are less competitive when they think their partner is just a machine. I'm concerned that this could have contributed to the finding that the collaboration of participants with S# matches their collaboration with other human participants.

Author response: This is an important point we had not previously considered. In the revised manuscript, we now provide an additional analysis exploring this question in Section F.3.3 of the SI. The rate of mutual cooperation with S# was 59% when a participant identified it as a human, and 62% when a participant identified it as a machine, and this small difference is not statistically significant. Accordingly, variations in participants' responses to the Turing test do not seem to impact performance.

Second, Figure 3(a) shows that S#, compared to humans, used more affectively charged negative messages--more hateful, threatening, and less praiseful. Does the use of such messages predict performance? Does this depend on whether the partner is a machine (or thought to be a machine) rather than a human being? And if there is such dependency, does it explain the finding that the collaboration of participants with S# matches their collaboration with other human participants?

Author response: To address this point, we now report additional analyses (see also the response to reviewer 1). These analyses show that the greater use of Hate and Threats messages by S# (and its lesser use of Praise messages) is unlikely to have increased cooperation because the use of Hate and Threat message show a strong *negative* correlation of -0.80 with cooperation (and the use of Praise messages shows a small positive correlation with cooperation). This relation is not moderated by whether the partner is thought to be human or machine. In other words, S# is more likely than humans to send negative messages when its partner does not cooperate (hence the strong negative correlation), but the users do not seem to react differently to these messages as a function of whether they believe their partner to be human or machine. This additional analysis is provided in Section F.3.3 in the SI, and is references in the revised caption of Figure 3.

Overall, I learned a lot from this paper and think that a revision that addresses these would be a good candidate for publication in this journal.

Reviewer #3 (Remarks to the Author):

The paper considers a very interesting and important problem.

However, at its current state it could not be accepted to Nature Communications.

The paper is not well written. Its main contribution is a modified algorithm for S++, called S#, aimed at generating non-binding, costless communication with people for establishing relationships.

S++ is only described on page 13, and S# on page 16. It is hard to understand anything that is presented before that as many terms are used without proper definitions or examples.

Author response: We thank the reviewer for highlighting the need to improve the presentation of the paper. We have now included more explicit pointers to the Methods and Supplementary Information (appendix) in the introduction and the results section so that the reader knows where to look for details as they read the overview of the main results. In a sense, the sections of the paper could be read in the following order to provide a more traditional flow for some disciplines: Introduction, methods (section 3), results (section 1), discussion (section 2). We hope we have found a way to follow the reviewer's suggestion while also complying with Nature Communications' formatting guidelines, which requires placing details of the methods at the end.

Surprisingly, the results are given without the full experimental details which were omitted "in the interest of space".

Author response: We have only taken space into account to limit results reported in the main paper. Please note that the full detailed experimental design and statistical analysis are not omitted, but rather placed in the Supplementary Information (appendix). In the revised manuscript, we also report additional hypothesis testing conducted in response to reviewers 1 and 2. We believe the experimental evaluation reported in the main paper and the appendix is now quite extensive.

The results seem significant but it's hard to understand exactly what is the significance of this paper as the modification of S++ is marginal and the experimental results are reported for only five, very simple repeated game settings. I did like the extensive comparison with other algorithms.

Author response:

Re: Contribution: We thank the reviewer for prompting us to more clearly state the primary contributions and significance of the paper in the introduction. We have now revised the paper to state the significance of our findings more explicitly upfront. The revised introduction now lists explicitly the paper's primary contributions. In fact, we borrowed some language from Reviewer 1, which captured these contributions well. In particular, we modified the last paragraph of the introduction to read as follows:

“The primary contribution of this work is three-fold. First, we conduct an extensive comparison of existing algorithms for repeated games. This evaluation is summarized in Methods (Section 3.2) and detailed in full in the SI (SI.B). Second, we develop and analyze a new learning algorithm that couples a state-of-the-art machine-learning algorithm (the highest performing algorithm in our comparisons of algorithms) with novel mechanisms for generating and responding to signals at levels conducive to human understanding. This algorithm is overviewed in Methods (Sections 3.3-3.4) and described in detail in the SI (SI.C). Finally, via extensive simulations and user studies, we show that this learning algorithm learns to establish and maintain effective relationships with people and other machines in a wide-variety of repeated stochastic games at levels that rival human cooperation, a feat not achieved by prior algorithms. In so doing, we investigate the algorithmic mechanisms that are responsible for the algorithm's success. These results are summarized and discussed in the next section, and given in full detail in the SI (SI.D-F).”

Re: Number of games: Our comparison of existing algorithms was conducted across 720 different games, which provides an extensive test of the generality of the algorithms. For user studies, we cannot cast as broad of a net, so we carefully selected the games we tested so that they represented different *payoff families*. Even so, we believe the resulting analysis is quite extensive compared to the extent of games analyzed in other published works. We also hope that the present paper will encourage other scientists to expand the scope of human-machine cooperation experiments, to be more in line with the thousands of scholarly articles written about human-human game play.

Re: Comparison of algorithms: The comparison of existing algorithms took a lot of effort and careful design, so we are glad that it resonated with the reviewer.

REVIEWERS' COMMENTS:

Reviewer #1 (Remarks to the Author):

All my previous feedback has now been taken into account and I am happy for the manuscript to be published in the revised form.

Reviewer #2 (Remarks to the Author):

The authors adequately addressed my questions. I have no further comments.